# Evaluation of neonatal mortality data completeness and accuracy in Ghana

**Dora Dadzie**[1][◎], **Richard Okyere Boadu**[2][◎], **Cyril Mark Engmann**[3,4,5], **Nana Amma Yeboaa Twum-Danso**[6,7][◎] *

1 Cape Coast Teaching Hospital, Cape Coast, Ghana, 2 Department of Health Information Management, University of Cape Coast, Cape Coast, Ghana, 3 Maternal, Newborn and Child Health and Nutrition, PATH, Seattle, WA, United States of America, 4 Department of Paediatrics, University of Washington School of Medicine, Seattle, WA, United States of America, 5 Department of Global Health, University of Washington School of Public Health, Seattle, WA, United States of America, 6 TD Health, Accra, Ghana, 7 Gillings School of Global Public Health, University of North Carolina, Chapel Hill, NC, United States of America

◎ These authors contributed equally to this work.
* ntwumdanso@gmail.com

**Data Availability Statement:** All relevant data are within the paper and its Supporting Information files.

**Funding:** The study was funded by the Children's Investment Fund Foundation with project code R-

## Abstract

### Background

Cause-specific mortality data are required to set interventions to reduce neonatal mortality. However, in many developing countries, these data are either lacking or of low quality. We assessed the completeness and accuracy of cause of death (COD) data for neonates in Ghana to assess their usability for monitoring the effectiveness of health system interventions aimed at improving neonatal survival.

### Methods

A lot quality assurance sampling survey was conducted in 20 hospitals in the public sector across four regions of Ghana. Institutional neonatal deaths (IND) occurring from 2014 through 2017 were divided into lots, defined as neonatal deaths occurring in a selected facility in a calendar year. A total of 52 eligible lots were selected: 10 from Ashanti region, and 14 each from Brong Ahafo, Eastern and Volta region. Nine lots were from 2014, 11 from 2015 and 16 each were from 2016 and 2017. The cause of death (COD) of 20 IND per lot were abstracted from admission and discharge (A&D) registers and validated against the COD recorded in death certificates, clinician's notes or neonatal death audit reports for consistency. With the error threshold set at 5%, $\geq$ 17 correctly matched diagnoses in a sample of 20 deaths would make the lot accurate for COD diagnosis. Completeness of COD data was measured by calculating the proportion of IND that had death certificates completed.

### Results

Nineteen out of 52 eligible (36.5%) lots had accurate COD diagnoses recorded in their A&D registers. The regional distribution of lots with accurate COD data is as follows: Ashanti (4, 21.2%), Brong Ahafo (7, 36.8%), Eastern (4, 21.1%) and Volta (4, 21.1%). Majority (9, 47.4%) of lots with accurate data were from 2016, followed by 2015 and 2017 with four

1703-01827-TD Health. The funders had no role in study design, data collection and analysis, decision to publish, or preparation of the manuscript.

**Competing interests:** The authors have declared that no competing interests exist.

(21.1%) lots. Two (10.5%) lots had accurate COD data in 2014. Only 22% (239/1040) of sampled IND had completed death certificates.

## Conclusion

Death certificates were not reliably completed for IND in a sample of health facilities in Ghana from 2014 through 2017. The accuracy of cause-specific mortality data recorded in A&D registers was also below the desired target. Thus, recorded IND data in public sector health facilities in Ghana are not valid enough for decision-making or planning. Periodic data quality assessments can determine the magnitude of the data quality concerns and guide site-specific improvements in mortality data management.

## Introduction

Data on cause-specific neonatal morbidity and mortality are needed to guide policies and interventions aimed at improving the lives of neonates at both global and local levels [1–3]. They are also needed to monitor the effectiveness of interventions aimed at neonatal mortality reduction [4, 5]. Knowledge of the contributors to the neonatal death burden is particularly relevant for planning interventions to meet the targets set in the third sustainable development goal (SDG 3) [6, 7]. Globally, about 2.6 million neonates die annually, representing 46% of all under-five child deaths [8, 9]. Majority of these deaths occur in developing countries [9]. Although globally, under-five mortality rate has declined remarkably since the 1990s, neonatal mortality rate has declined at a much slower rate [10, 11]. If current trends were to remain the same, the SDG 3 target of reducing neonatal mortality rate to 12 per 1,000 live births by 2030, and the subsequent reduction in under-five mortality, will not be achieved [6]. This underpins the demand for the scale up of interventions accompanied by effective implementation to accelerate the reduction. For interventions to be effective and optimal however, they need to be evidence-based and targeted to local needs; to a great extent, this is dependent on the availability of quality data needed for planning, monitoring, evaluation and redesign as needed. In developing countries which bear the brunt of neonatal deaths, there are inadequate vital registration systems [12–14] with poor quality of cause of death (COD) data [15–17]. Mortality estimates in these countries are therefore traditionally derived from modelled data from other countries with better quality data [1, 15, 18]. Consequently, the estimated rates might not truly reflect the neonatal mortality burden.

Ghana, like other developing countries has experienced a much smaller reduction in neonatal mortality, reducing by 42% from 43 per 1,000 live births in 1988 to 25 per 1,000 live births in 2017, compared to under-five mortality which reduced by 67% from 155 per 1,000 live births in 1988 to 52 per 1,000 live births in 2017 [19]. In fact, between 2007 and 2018, Ghana's neonatal mortality rate reduced by only 14% from 29 to 25 deaths per 1,000 live births [19]. The key sources of mortality data in Ghana are vital registration systems, population-level surveys, and facility-based medical records [20]. However, less than a third of all deaths in Ghana are reported to the vital registration system, with even lower proportions reported for infant deaths [21]. Also, nationwide demographic and health surveys are conducted only periodically, making facility-based medical records the more regularly available source of mortality data in the country. Facility-based sources of neonatal COD data include admission and discharge (A&D) registers, clinicians' notes, neonatal death audit reports, and neonatal death certificates. On initial presentation to a health facility, neonates requiring admission are assigned folders

in which clinicians' notes are documented. The initial diagnoses on presentation are recorded in the A&D registers, which are supposed to be updated with the final diagnoses upon discharge or death. Medical COD certificates are expected to be issued for all deaths, as stipulated in the Registration of Births and Deaths Act, 1965 (ACT 301) of Ghana [22]. Monthly summary data of COD abstracted from the A&D registers and death certificates are fed into the national health information database. The medical COD certificates are also presented to the Births and Death Registry for registration. Concerns about the quality of data are valid due to the passive nature of data collection procedures, as routine health data have been found in many instances to be of low accuracy [23–26], low reliability [25], incomplete [25, 27], and delayed [28]. To the best of our knowledge, there is no available information on the quality of mortality-specific data of neonates recorded in medical records in Ghana. A few studies have assessed the errors made during completion of medical certificates [29, 30], but none has assessed their accessibility as a source of neonatal mortality data. Moreover, whether the COD information recorded in these sources accurately represent the true burden is not known. The goal of this study was to validate medical records as a credible source of neonatal mortality information in Ghana. Our primary objective was to assess the accuracy of COD data for neonates recorded in A&D registers from 2014 through 2017 while our secondary objective was to assess the proportion of neonatal deaths for which death certificates were issued during this period.

## Methods

### Context

The Ghana Making Every Baby Count Initiative (MEBCI) was a five-year project which sought to improve and standardize neonatal care in support of the Ghana government's goal to accelerate the reduction of neonatal mortality from 32 per 1,000 livebirths in 2011 to 21 per 1,000 livebirths in 2018. MEBCI approached this aim with specific interventions at national level, the Greater Accra Regional Hospital, the largest regional hospital in the country, as well as the targeted administrative regions of Ashanti, Brong Ahafo, Eastern and Volta. The project was a partnership across the Ghana Health Service (GHS), Kybele Inc. and PATH, and was funded by the Children's Investment Fund Foundation (CIFF) from September 2013 to August 2018. MEBCI's goal was that, by 2018, 90% of neonates delivered in health care facilities in targeted regions would receive essential neonatal care and appropriate interventions to address asphyxia, infection, and prematurity per government guidelines. One of the components of the evaluation of the MEBCI project included a temporal assessment of the institutional neonatal death (IND) rates and case fatality rates. To this end, data on causes of neonatal death were abstracted from A&D registers from 155 health facilities that benefited directly from MEBCI. During preliminary analysis of the COD data, concerns about some listed neonatal COD diagnoses such as HIV infection and Hepatitis B infection arose. This study was undertaken due to concerns about the quality of the data following this preliminary analysis.

### Sample size and sampling

A Lot Quality Assurance Sampling (LQAS) survey was conducted in 20 government and quasi-government health facilities in Ashanti, Brong Ahafo, Eastern and Volta regions. The LQAS, a type of sampling and analytic method, is underpinned by the assumption that a random sample as small as 19 would provide a statistically reasonable inference comparable to that from a large sample size, about whether a pre-determined target for an indicator has been achieved or not achieved [31, 32]. The LQAS method comprises three steps. Firstly, sub-populations, termed 'lots' within the study population are defined and a random small sample is

taken from each lot. The lots are then tested separately using the decision rule to determine whether they have achieved the target for the indicator being evaluated. If the number of 'errors' in the sample does not exceed the threshold for permissible 'errors', then the decision is made that the target has been achieved [31, 32]. Lastly, the lots are weighted and aggregated to provide regional-level estimates for the indicator. The LQAS methodology, though traditionally used in industry for quality control, has been adopted and used extensively in data quality assessment [32], epidemiologic surveillance [33], and health programme monitoring [34, 35].

For the period under review, there were a total of 5,814 INDs recorded in the full census of 155 health facilities. A multi-stage sampling technique was used to sample IND. Health facilities in each of the four administrative regions were selected based on their IND burden. The total number of neonatal deaths occurring in each facility from 2014 through 2017 was determined. The facilities were then ranked and for each administrative region, the top 20% high burden facilities or high-burden facilities cumulatively contributing at least 80% of deaths (whichever criterion was met first) were selected for the survey. At the facility level, IND were divided into lots. A lot was defined as containing all neonatal deaths within a calendar year (i.e. 2014, 2015, 2016, and 2017) in the facility. In each lot, 20 IND were sampled from the A&D registers using a systematic random sampling approach. A sample size of 20 was expected to provide an acceptable margin of error and the same statistical precision, as would a sample size greater than 20. The folder numbers of all IND cases in a lot were recorded and assigned numbers (from one to 'n', where n is the number for the last case in the calendar year). The total number 'n' was divided by 20 to obtain the sample interval, 'k'. A random number generator app was used to randomly select the first IND case, and subsequent 'k$^{th}$' IND cases were selected until 20 IND cases per lot were obtained. Lots which had less than the minimum sample of 20 neonatal deaths were excluded from the survey.

## Data collection

Data abstraction was done by a public health physician with clinical experience in the Ghanaian health system and thus was knowledgeable about the processes and documents involved in neonatal health care. For each sampled IND case, the final diagnosis recorded in the A&D register was abstracted. If no final diagnosis was recorded in the A&D register, the provisional diagnosis was abstracted instead. The medical COD certificate issued for that IND was traced and the principal COD as transcribed in the death certificate was also abstracted. If no medical COD certificate was issued, the final diagnosis or COD recorded in the clinician's notes or neonatal death audit record was abstracted as shown in the flowchart (**Fig 1**). For the analysis, the gold standard was defined as the medical COD certificate. Thus, the final diagnosis recorded in the A&D registers (or provisional diagnosis if no final diagnosis recorded) was compared with that recorded in the medical COD certificate. If no certificate was issued, the diagnosis was compared with that in either the clinician's notes or neonatal death audit record.

## Data analysis

The diagnosis in the A&D register was said to be accurately matched to that in the validation source if the two diagnoses were consistent. Within a lot of twenty sampled IND (n = 20) with $X_i$ accurately matched diagnoses and decision rule 'd', if $X_i \geq d$, then the lot was classified as having accurate COD data. If $X_i < d$, then the lot would have failed to achieve the target of having accurate COD data. The threshold for accurate data was set at 95%. According to the Lemeshow and Taber's LQAS table, for a target of 95% and sample size of 20, the decision rule

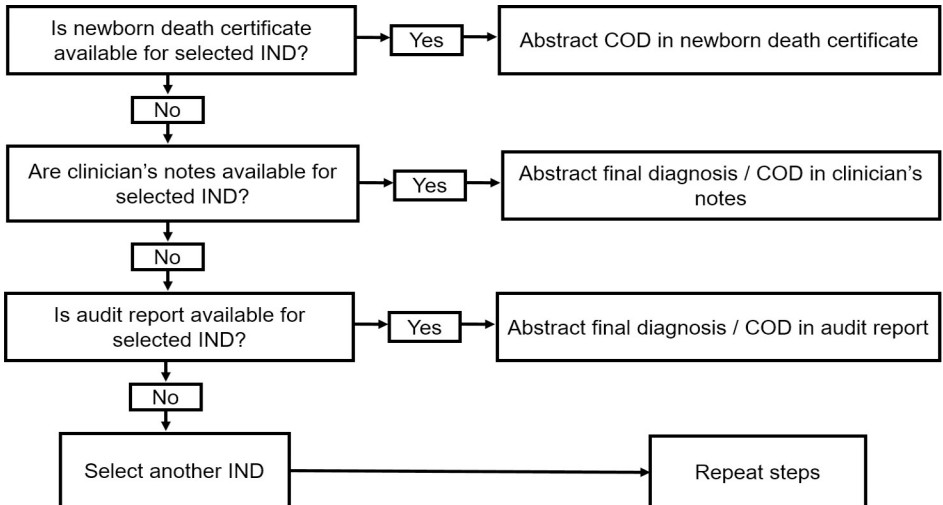

**Fig 1. Flowchart for data abstraction from mortality data sources (COD: Cause of death).**

'd' is 17, meaning a lot would have a minimum of 17 accurately matched diagnoses to have accurate COD data.

Decision rule: *If $X_i \geq 17$, the lot has accurate cause of neonatal death data; $X_i < d$, the lot does not have accurate cause of neonatal death data.*

Data across the lots were aggregated to provide overall, regional-level and yearly weighted estimates of data accuracy. Weighting was necessary to adjust for the differences in total facility and yearly IND burden and was done using the direct adjustment method. Lots were weighted according to their proportionate contribution to the total IND burden. For each lot, the proportion of accurately matched diagnoses was multiplied by the weight (number of IND for the lot divided by the total number of IND) to obtain the weighted proportion of accurately matched diagnoses for the lot. The overall, regional-level and yearly weighted accuracy estimates were then calculated by summing up the individual weighted estimates of involved lots.

### Ethical considerations

Approval for the independent evaluation of MEBCI was granted by GHS Ethical Review Approval Board on 18th August 2017, with identification number GHS-ERC 20/06/17. Permission was obtained from the Director General of the GHS as well as the regional health directorates and health facilities before data collection commenced. Patient informed consent was waived by the ethics committee due to the nature of the evaluation. Patients' folder numbers were abstracted from admission and discharge registers only to be able to trace medical cause of death certificates, mortality audit reports or clinical notes. Following validation of cause of death diagnosis, folder numbers were replaced with numbers assigned by data abstractor in the study database. Only study authors have access to anonymized individual-level data.

## Results

### Characteristics of lots

Overall, 80 lots (20 per year) were sampled, out of which 52 (65%) had sufficient minimum neonatal deaths for the LQAS survey. Majority were from 2016 (16/52, 30.7%) and 2017 (16/52, 30.7%) (**Fig 2**). Ten (19.2%) out of the 52 lots were from facilities located in Ashanti region while Brong Ahafo, Eastern and Volta each had 14 (26.9%) lots from health facilities located in

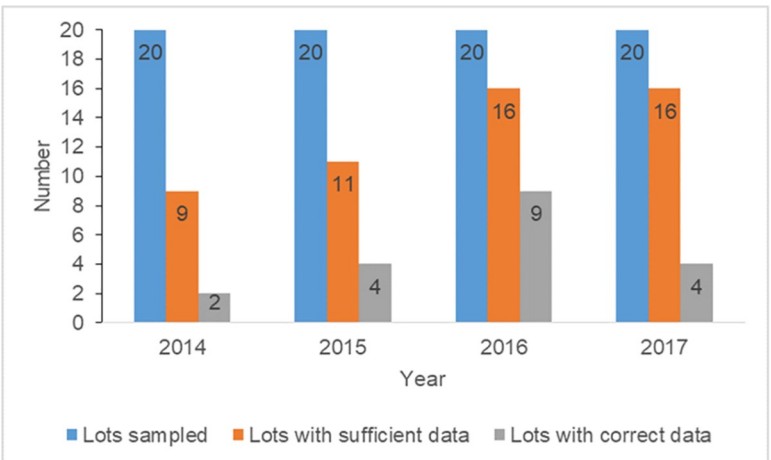

**Fig 2. Yearly distribution of lots with sufficient minimum data (≥ 20 neonatal deaths) and accurate data.**

these regions. About 40% (21/52) of selected lots were from government-owned facilities while the remainder (59.6% or 31/52) were from quasi-government facilities. A total of 1,040 neonatal deaths were obtained from the 52 lots.

## Availability of death certificates

Only 22.1% (230/1,040) of selected IND had death certificates issued, thus were validated using this gold standard. The remaining 77.9% of IND (810/1,040) were validated using clinician's notes or neonatal death audit records. In two regions (Ashanti and Volta), no neonatal death certificates were issued at all during the study period. Eastern region had the highest proportion (60.7%, 170/280) of neonatal deaths issued medical COD certificates. In the four regions, the cumulative proportion of deaths issued certificates was highest in 2014 (33.3%, 60/180) and lowest in 2015 (18.2%, 40/220). Fewer IND had death certificates issued in the government facilities (19.0% i.e. 80/420) than in the quasi-government facilities (24.2% i.e. 150/620) (**Table 1**).

## Data accuracy

Overall, 75.9% (95% CI: 74.5–77.3) of neonatal COD data recorded in A&D registers from the four regions were accurately matched to the validation sources. The proportion was highest in Brong Ahafo region (82.2%, 95% CI: 79.9–84.5) and lowest in Volta region (62.5%, 95% CI: 58.9–66.0). The proportion of accurately matched data was also highest in 2016 (79.3%, 95% CI: 76.6–81.8) and lowest in 2017 (72.4%, 95% CI: 69.6–74.9). There were slightly more accurately matched diagnoses in quasi-government facilities (78.6%, 95% CI: 76.8–80.3), than in government facilities (72.6% 95% CI: 70.5–74.7) (**Table 1**). Examples of common accurately and inaccurately recorded COD in A&D registers, juxtaposed with COD recorded in the gold standard source document for the same neonate are presented in **Table 2** for illustration purposes.

Of the 52 lots, a total of 19 (28.8%) met the target of ≥ 17/20 accurately matched diagnoses. There were regional variations, with only one region (Brong Ahafo) having about half of its lots meeting the desired target. Except for 2016, which had a little more than 50% of lots meeting the target, other years had less than 50% of lots meeting the target (**Table 1**). There were small differences in the proportion of lots meeting the target between government (38.1%, 8/21) and quasi-government hospitals (35.5%, 11/31).

**Table 1. Death certificate completion and accuracy of cause-specific neonatal mortality data in a sample of health facilities.**

| Variable | % IND with death certificate completed | Number of eligible lots | Number (%) of lots with accurate data | % estimate data accuracy (95% CI) |
|---|---|---|---|---|
| Region | | | | |
| Ashanti | 0% (0/200) | 10 | 4 (40.0%) | 74.7 (70.9–78.4) |
| Brong Ahafo | 21.4% (60/280) | 14 | 7 (50.5%) | 82.2 (79.9–84.5) |
| Eastern | 60.7% (170/280) | 14 | 4 (28.6%) | 78.2 (76.0–80.4) |
| Volta | 0% (0/280) | 14 | 4 (28.6%) | 62.5 (58.9–66.0) |
| Year | | | | |
| 2014 | 33.3% (60/180) | 9 | 2 (22.2%) | 77.9 (74.8–81.0) |
| 2015 | 18.2% (40/220) | 11 | 4 (36.4%) | 74.4 (71.3–77.4) |
| 2016 | 19.1% (61/320) | 11 | 9 (56.3%) | 79.3 (76.9–81.8) |
| 2017 | 21.6% (69/320) | 16 | 4 (25.0%) | 72.4 (69.8–74.9) |
| Facility ownership | | | | |
| Government | 19.0% (80/420) | 21 | 8 (38.1%) | 72.6 (70.5–74.7) |
| Quasi-government | 24.2% (150/620) | 31 | 11 (35.5%) | 78.6 (76.8–80.3) |
| Total | 22.1% (230/1,040) | 52 | 19 (36.5%) | 75.9 (74.5–77.3) |

## Discussion

We assessed the accuracy of neonatal COD data recorded in health facility A&D registers across four administrative regions of Ghana from 2014 through 2017. We found that less than 25% of IND were accompanied by a medical death certificate, the gold standard, and thus had to be validated by other sources. Moreover, two of the four regions assessed issued zero medical death certificates for IND during the period under review. The overall accuracy of the 80 lots sampled was estimated at 76% though there were wide variations across calendar years, administrative regions and type of facility ownership.

The finding of low accuracy is not unexpected as studies in other developing countries report similar results [17, 23, 24, 32, 36, 37]. It is however unclear why health facilities in some selected regions had comparatively bigger proportions of lots meeting the target than the others, or why data accuracy varied over the years. The documentation of accurate data is influenced by several factors that include staff motivation and perceived usefulness of the data [38]. If data are deemed useful, there is the tendency to capture accurate information and all of its components [38]. The importance of accurate COD data is rooted in its implications should the wrong information be used to guide priorities, policies or interventions at the local or

**Table 2. Examples of common accurately and inaccurately matched recorded cause of death diagnosis.**

| Cause of death recorded in A&D register for neonate | Cause of death recorded in gold standard source document for same neonate |
|---|---|
| Extreme prematurity | Extreme prematurity |
| Neonatal sepsis | Neonatal sepsis |
| Neonatal asphyxia | Neonatal asphyxia |
| Neonatal jaundice | Neonatal jaundice |
| Respiratory distress | Congenital heart disease |
| Home delivery | Neonatal sepsis |
| Respiratory distress | Neonatal sepsis |
| HIV exposed baby | Severe birth asphyxia |
| Intracranial space occupying lesion | Neonatal tetanus |

national level. Neonatal interventions are targeted at different periods during pre-pregnancy, pregnancy, delivery and the postnatal period depending on the disease of interest. For example, interventions for asphyxia may be targeted for the intrapartum and immediate postnatal period, whilst those for some congenital malformations may be most effective during early pregnancy or before conception. Due to the cost associated with implementing such interventions at large scale, data-driven decision-making processes are critical to monitor the interventions, optimise their impact and prevent wastage, especially in resource-constrained settings such as Ghana [2, 8].

A&D registers are important original sources of routine health surveillance data. Although we assessed only the accuracy of mortality data, it is possible that morbidity data which share the same source documents would equally be of low accuracy. Data from registers are eventually fed into the national health information database and used to inform policies, monitor interventions and contribute to the literature on neonatal mortality. Against the background of poor vital registration in developing countries, hospital registers remain essential alternate sources of COD data [12, 13, 20]. In neonates, many diseases interact, and it is advantageous to know the specific complications contributing to neonatal deaths. For example, in one sampled facility, it was observed that neonatal deaths attributable to birth asphyxia, as recorded in A&D registers, were actually due to congenital heart diseases. Many such scenarios were detected in preterm neonates dying from sepsis or aspiration pneumonia rather than from direct complications of prematurity. These findings undermine the usefulness of registers as credible sources of morbidity and mortality information and reduce practitioners' and policy-makers' ability to make decisions based on local real-time data.

A key component of data quality is data accessibility, which is assessed by the availability of the required information for action. The Registration of Birth and Deaths Act (ACT) 301 implemented in 1965 outlines the procedure for death registration, which begins with the last attending medical practitioner providing a COD for all deaths occurring in health facilities, including foetal deaths [21]. Medical COD certificates also serve many purposes other than for death registration. They are an important source of routine cause of death data and may be used to determine the leading causes of death for setting health interventions. They may also be required as proof of death for claim to life insurance benefits and inheritance. Therefore, their low issuance is very disconcerting as it does not conform to the legal directive by the Registry of Births and Deaths Act. Also, using these incomplete, thus, unreliable data to set interventions aimed at reducing neonatal mortality can potentially mislead decision makers to set inappropriate interventions, which may consequently fail to achieve the intended targets. In Ghana, although no study has assessed the proportion of deaths issued death certificates to our knowledge, their demand before issuing burial permits for burials in public cemeteries as well for medico-legal reasons (e.g. for inheritance claims) ensures the certification of many adult deaths. However, for neonates who are mostly buried on family lands, this procedure is not duly followed. Moreover, the demand for a certificate for neonatal death is low due to its perceived insignificance as a legal document and medical practitioners not knowing of its legal backing.

The LQAS method has increasingly been accepted as a quick and less expensive alternative to traditional surveys, and has been employed in monitoring by programmes, health surveys and data quality assessments [39–45]. The primary objective of LQAS in this data accuracy assessment was to categorize sampled facilities which had accurate data and which did not, using a 5% error margin. Grouping them would make it easier to determine which facilities required an intervention. Although not the primary aim, we also aggregated data from the lots to provide an estimate of the level of data accuracy. We found the level of data accuracy for the four regions, and for each region to be similar to values found in other African countries [23, 24, 37].

## Limitations

Although the different data sources for validation might have different levels of precision regarding the cause of death or diagnosis, the underlying principal diagnoses were expected to remain the same irrespective of the source. Only few facilities were sampled from each region. However, they were presumed representative of the regions as they contributed the greatest proportions to the neonatal death burden across the facilities that benefited from MEBCI. Despite the small sample size that was used, we expected that estimates derived would provide statistically reasonable inferences, similar to those derived from large sample surveys.

## Conclusion

COD certificates were not reliably completed for neonatal deaths from 2014 through 2017 in major health facilities across four regions in Ghana. The COD or final diagnosis data recorded A&D registers in many of these facilities were found to be inaccurate. Both of these findings limit the ability to monitor the effectiveness of health system interventions and guide programmatic and policy decisions needed to accelerate the reduction of neonatal deaths and contribute towards the achievement of SDG3. We recommend to the Ghana Health Service to include a review of neonatal death certificates in neonatal death audits to ensure it gets completed. Periodic data quality assessments can determine the magnitude of the data quality concerns and guide site-specific improvements in data management.

## Supporting information

**S1 Dataset.**
(XLSX)

## Acknowledgments

The authors thank the Regional Health Directors, Regional Neonatal Focal Persons, and health staff of participating facilities for their assistance. We would like to express our gratitude to TD Health staff for all their support.

## Author Contributions

**Conceptualization:** Dora Dadzie, Richard Okyere Boadu, Nana Amma Yeboaa Twum-Danso.

**Data curation:** Dora Dadzie.

**Formal analysis:** Dora Dadzie, Richard Okyere Boadu, Nana Amma Yeboaa Twum-Danso.

**Funding acquisition:** Nana Amma Yeboaa Twum-Danso.

**Methodology:** Dora Dadzie, Richard Okyere Boadu, Nana Amma Yeboaa Twum-Danso.

**Supervision:** Nana Amma Yeboaa Twum-Danso.

**Writing – original draft:** Dora Dadzie.

**Writing – review & editing:** Dora Dadzie, Richard Okyere Boadu, Cyril Mark Engmann, Nana Amma Yeboaa Twum-Danso.

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
