## [Decision Letter · Decision Letter 0]

22 Jun 2020

PONE-D-20-01438

Evaluating the completeness and accuracy of cause of institutional neonatal death data in Ghana using lot quality assurance sampling

PLOS ONE

Dear Dr. %LAST_NAME%,

Thank you for submitting your manuscript to PLOS ONE. After careful consideration, we feel that it has merit but does not fully meet PLOS ONE’s publication criteria as it currently stands. Therefore, we invite you to submit a revised version of the manuscript that addresses the points raised during the review process.

Please note that all suggestions must be faced and listed with the response following it. If any suggestion was not accepted, please justify it convincingly.

We look forward to receiving your revised manuscript.

Kind regards,

Ricardo Q. Gurgel, PhD

Academic Editor

PLOS ONE

Journal Requirements:

2. ‘In ethics statement in the manuscript and in the online submission form, please provide additional information about the patient records used in your retrospective study. Specifically, please ensure that you have discussed whether all data were fully anonymized before you accessed them and/or whether the IRB or ethics committee waived the requirement for informed consent. If patients provided informed written consent to have data from their medical records used in research, please include this information.

3. We note that Figure 1 in your submission contain map images which may be copyrighted. All PLOS content is published under the Creative Commons Attribution License (CC BY 4.0), which means that the manuscript, images, and Supporting Information files will be freely available online, and any third party is permitted to access, download, copy, distribute, and use these materials in any way, even commercially, with proper attribution. For these reasons, we cannot publish previously copyrighted maps or satellite images created using proprietary data, such as Google software (Google Maps, Street View, and Earth). For more information, see our copyright guidelines: http://journals.plos.org/plosone/s/licenses-and-copyright.

3.1.    You may seek permission from the original copyright holder of Figure 1 to publish the content specifically under the CC BY 4.0 license.

3.2.    If you are unable to obtain permission from the original copyright holder to publish these figures under the CC BY 4.0 license or if the copyright holder’s requirements are incompatible with the CC BY 4.0 license, please either i) remove the figure or ii) supply a replacement figure that complies with the CC BY 4.0 license. Please check copyright information on all replacement figures and update the figure caption with source information. If applicable, please specify in the figure caption text when a figure is similar but not identical to the original image and is therefore for illustrative purposes only.

Reviewers' comments:

Reviewer's Responses to Questions

**Comments to the Author**

1. Is the manuscript technically sound, and do the data support the conclusions?

Reviewer #1: Yes

Reviewer #2: Partly

2. Has the statistical analysis been performed appropriately and rigorously?

Reviewer #1: Yes

Reviewer #2: Yes

3. Have the authors made all data underlying the findings in their manuscript fully available?

Reviewer #1: Yes

Reviewer #2: Yes

4. Is the manuscript presented in an intelligible fashion and written in standard English?

Reviewer #1: Yes

Reviewer #2: Yes 

5. Review Comments to the Author

Reviewer #1: Line 3 : Short title seems better

Line 23 : Language correction ; mentioned as sticky note

Line 104, method section : To omit cost analysis as was not reflected in method section on how it was done and in result section also

Line 170 : Table 1 ; to omit content in bracket in column 5 as information is given in method section

Line 216 : To omit the line as it was not objective/ result findings

Reviewer #2: Dear Authors,

I have reviewed your manuscript and read with a great interest. The application of LQAS technique for evaluating the completeness and accuracy of cause of institutional neonatal death data in Ghana is a relevant technique that facilitate rapid and statistically valid data quality assessment in SSA countries. Below are some comments which I think, if considered would strengthen your manuscript.

Abstract

Please consider separating the "Methods" from "Findings", and name the later as "Results".

Methods: provide a brief description about the number of lights included in the study. In this description, be specific about number of lots by facility, year and total (overall). This will ensure easiness of understanding your results. Please be specific also about the measure of completeness of COD data.

Results: only summarise the key findings, i.e., related to completeness and accuracy of causes of IND data. For example, "Nineteen out of 52 (36.5%) eligible lots met the target for accurate COD." How many of these were accurate? If the answer is 76%, then this sentence should have come before reporting accuracy results.

Conclusion: are the death certificates the source of routine data used for policy and administrative decision making? The conclusion both in the abstract and the main text started with indicating that they were not reliably completed. What would be the implications for this?

Introduction

Line 64-65, specific the number of neaonatal and U5 deaths reduced from what specific values between the specified years. Consider also splitting the sentence in two parts where you started with "in fact..." on line 66.

Methods

Context: Line 103-104, in your sentence you mentioned "...qualitative and quantitative health". Please rephrase this to make it easier to understand. The first paragraph showing the context provided a brief background or the motivation for this study. However, in the discussion, have you considered comparing your results to those reported by the MEBCI project? What data quality issues did they report compared to your findings? In this paragraph, consider also specifying the initial data quality concerns/ issues observed or reported before this study. Also, did the MEBCI publish any paper relating such data quality issues? If so, incorporate such findings as part of the context section.

Line 121: A multistage sampling technique was used to sample what?

Data collection and analysis: Consider splitting data collection and data analysis sections. Also, to enable understanding the LQAS methodology, I recommend you have a section of "Sample size and Sampling" where the second paragraph of the context section will fit better. Within this new section you also provide the formula used for sample size calculation (if used) as well as the threshold value(s) as per the LQAS theory. This is very important to readers who never heard of this methodology. After providing the sample size information, then describe the assumptions of LQAS and how it was applied as described in the first paragraph of data collection and analysis.

The information on the second paragraph of data collection and analysis section can go to the section describing "Data collection".

Line 146-148: Data accuracy was set at 95% and further described as 17/20 correct match to mean accuracy. Someone would imply 17/20 as 85% instead of 95%. To avoid confusion, please rephrase this sentence and provide a more clear description of the value of accuracy used in relation of 17/20 correct matches.

Line 148-149: You stated that. "Data from the lots were aggregated to provide weighted estimates of accuracy for the regions and years." Did you actually weight the data? How was this done? Provide a description in the data analysis section.

Results

For all figure citations within the text, I think the journal recommends citing as Fig 1. Fig 2

The descriptions related to Table 1 are summary estimates. I recommend you combine Table 1 and 2, i.e., Table 1 to be summarised in Table 2. Otherwise, summarise the information about the number of lots in Table 2 for ease of reference with the number of lots with accurate data. It is important to ensure such descriptions are reflected in your table(s).

The columns in Table 2 are about validated with death certificates and clinician’s note or audit report. However, the title is about "Death certificate completion..." which I think does not reflect the content within this table. Please consider rephrasing to reflect the actual information. Likewise, lines 173-180 the descriptions are more on the death certificates issued while the results in the table are about the percentages (and numbers) of IND validated against the source documents. Please revise.

The last column of table 2, how were these percentages calculated. Please describe that in the methods section (data analysis) and/ or below this table.

Table 3: Consider including the most common causes of IND that were correctly matched. The reason for this is because, it would be even more informative to understand the common causes of neonatal deaths in Ghanaian health facilities. Correct matches is also a part of data quality (i.e., high accurate data). So, I think these should also be reported.

Please cite the respective table number where the results which are below Table 3 are described.

Discussion

The sentence in Line 242-243 "Medical COD certificates also serve many purposes other than for death registration." Specify the purposes for Medical COD certificates. This would also justify your conclusions. Please consider citing other literature (if any) that may support such purposes.

Conclusion

Based on the paragraph in line 226, A&D registers are important original source of routine health surveillance data. In your study, COD certificates were one of the reference documents to verify the causes of death data and therefore used to measure/ determine accuracy. My question is, why conclude COD certificates as unreliable, while also their data are not entered in the national system? Please consider this and revise your conclusion also in the abstract.

References

Please verify and correct some of your references such as number 27, 37 and 38.

6. PLOS authors have the option to publish the peer review history of their article (what does this mean?). If published, this will include your full peer review and any attached files.

**Do you want your identity to be public for this peer review?** For information about this choice, including consent withdrawal, please see our Privacy Policy.

Reviewer #1: Yes: Dr Sanjoy Kumer Dey

Reviewer #2: No

---

## [Author Response · Author response to Decision Letter 0]

2 Aug 2020

Response to Editor and Reviewers

Editor’s comments

1. Please ensure that your manuscript meets PLOS ONE's style requirements, including those for file naming. The PLOS ONE style templates can be found athttps://journals.plos.org/plosone/s/file?id=wjVg/PLOSOne_formatting_sample_main_body.pdf and https://journals.plos.org/plosone/s/file?id=ba62/PLOSOne_formatting_sample_title_authors_affiliations.pdf

Response: We have ensured that all files meet the PLOS ONE requirements.

2. ‘In ethics statement in the manuscript and in the online submission form, please provide additional information about the patient records used in your retrospective study. Specifically, please ensure that you have discussed whether all data were fully anonymized before you accessed them and/or whether the IRB or ethics committee waived the requirement for informed consent. If patients provided informed written consent to have data from their medical records used in research, please include this information.

Response: We thank the editor for drawing our attention to this. Patient informed consent was waived by the ethics committee due to the nature of the evaluation. Patients’ folder numbers were abstracted from admission and discharge registers only to be able to trace medical cause of death certificates, mortality audit reports or clinical notes. Following validation of cause of death diagnosis, folder numbers were replaced with numbers assigned by data abstractor in the study database. Only study authors have access to anonymized individual-level data. The ethics statement has been updated with this information. (Lines 173-180) 

3. We note that Figure 1 in your submission contain map images which may be copyrighted. All PLOS content is published under the Creative Commons Attribution License (CC BY 4.0), which means that the manuscript, images, and Supporting Information files will be freely available online, and any third party is permitted to access, download, copy, distribute, and use these materials in any way, even commercially, with proper attribution. For these reasons, we cannot publish previously copyrighted maps or satellite images created using proprietary data, such as Google software (Google Maps, Street View, and Earth). For more information, see our copyright guidelines: http://journals.plos.org/plosone/s/licenses-and-copyright.

3.1. You may seek permission from the original copyright holder of Figure 1 to publish the content specifically under the CC BY 4.0 license.

3.2. If you are unable to obtain permission from the original copyright holder to publish these figures under the CC BY 4.0 license or if the copyright holder’s requirements are incompatible with the CC BY 4.0 license, please either i) remove the figure or ii) supply a replacement figure that complies with the CC BY 4.0 license. Please check copyright information on all replacement figures and update the figure caption with source information. If applicable, please specify in the figure caption text when a figure is similar but not identical to the original image and is therefore for illustrative purposes only.

Response: The figure (formerly Figure 1) has been removed. 

 

Reviewer #1

Line 3: Short title seems better

Response: We thank the reviewer for the suggestion. We have changed the manuscript’s title from ‘Evaluating the completeness and accuracy of cause of institutional neonatal death data in Ghana using lot quality assurance sampling’ to ‘Evaluation of neonatal mortality data completeness and accuracy in Ghana’.

Line 23: Language correction; mentioned as sticky note

Response: We thank the reviewer for the correction. The sentenced has been corrected to read ‘Cause-specific mortality data are required to set interventions to reduce neonatal mortality.’ (Line 23) 

Line 104, method section: To omit cost analysis as was not reflected in method section on how it was done and in result section also

Response: We thank the reviewer for the comment. The sentence has been omitted. 

Line 170: Table 1; to omit content in bracket in column 5 as information is given in method section

Response: We are grateful to the reviewer for drawing our attention to this. The content has been omitted. 

Line 216: To omit the line as it was not objective/ result findings

Response: We are grateful to the reviewer for the suggestion. The sentence ‘With no incentive or feedback given to health staff, they may not be motivated to report quality data’ has been omitted. 

Reviewer #2 

Dear Authors, I have reviewed your manuscript and read with a great interest. The application of LQAS technique for evaluating the completeness and accuracy of cause of institutional neonatal death data in Ghana is a relevant technique that facilitate rapid and statistically valid data quality assessment in SSA countries. Below are some comments which I think, if considered would strengthen your manuscript.

Abstract

Please consider separating the "Methods" from "Findings", and name the later as "Results".

Response: We take note of the comment. We have separated the ‘Methods’ and ‘Findings’ sections of the abstract and renamed the latter as ‘Results’.

Methods: provide a brief description about the number of lights included in the study. In this description, be specific about number of lots by facility, year and total (overall). This will ensure easiness of understanding your results. Please be specific also about the measure of completeness of COD data.

Response: We take note of this instructive feedback. The ‘Methods’ section of the abstract has been revised to describe the distribution of the lots by year and region, and how data completeness was measured. The revised ‘Methods’ section is provided below.

‘A lot quality assurance sampling survey was conducted in 20 hospitals in the public sector across four regions of Ghana. Institutional neonatal deaths (IND) occurring from 2014 through 2017 were divided into lots, defined as neonatal deaths occurring in a selected facility in a calendar year. A total of 52 eligible lots were selected: 10 from Ashanti region, and 14 each from Brong Ahafo, Eastern and Volta region. Nine lots were from 2014, 11 from 2015 and 16 each were from 2016 and 2017. The cause of death (COD) of 20 IND per lot were abstracted from admission and discharge (A&D) registers and validated against the COD recorded in death certificates, clinician's notes or neonatal death audit reports for consistency. With the error threshold set at 5%, ≥ 17 correctly matched diagnoses in a sample of 20 deaths would make the lot accurate for COD diagnosis. Completeness of COD data was measured by calculating the proportion of IND that had death certificates completed.’ (Lines 27-36)

Results: Only summarise the key findings, i.e., related to completeness and accuracy of causes of IND data. For example, "Nineteen out of 52 (36.5%) eligible lots met the target for accurate COD." How many of these were accurate? If the answer is 76%, then this sentence should have come before reporting accuracy results.

Response: We thank the reviewer for the feedback. The ‘Results’ section of the abstract has been revised appropriately and reads as follows:

‘Nineteen out of 52 eligible (36.5%) lots had accurate COD diagnoses recorded in their A&D registers. The regional distribution of lots with accurate COD data is as follows: Ashanti (4, 21.2%), Brong Ahafo (7, 36.8%), Eastern (4, 21.1%) and Volta (4, 21.1%). Majority (9, 47.4%) of lots with accurate data were from 2016, followed by 2015 and 2017 with four (21.1%) lots. Two (10.5%) lots had accurate COD data in 2014. Only 22% (239/1040) of sampled IND had completed death certificates.’ (Lines 37-41)

Conclusion: Are the death certificates the source of routine data used for policy and administrative decision making? The conclusion both in the abstract and the main text started with indicating that they were not reliably completed. What would be the implications for this?

Response: Yes. Medical cause of death certificate is an important source of information on the leading causes of mortality for policy and administrative decision making in the country. Unreliable data on the causes of neonatal mortality can potentially mislead decision makers to set inappropriate interventions. This implication has been stated in the ‘Discussion’ section in the main text and is as follows: 

‘...Therefore, their low issuance is very disconcerting as it does not conform to the legal directive by the Registry of Births and Deaths Act. Also, using these incomplete, thus, unreliable data to set interventions aimed at reducing neonatal mortality can potentially mislead decision makers to set inappropriate interventions, which may consequently fail to achieve the intended targets.’ (Lines 264-268) 

Introduction 

Line 64-65, specific the number of neonatal and U5 deaths reduced from what specific values between the specified years. Consider also splitting the sentence in two parts where you started with "in fact..." on line 66.

Response: We are grateful to the reviewer for the comment. We have provided the specific numbers for neonatal and U5 deaths for the specified years. The sentence has been revised and reads as follows:

‘ Ghana, like other developing countries has experienced a much smaller reduction in neonatal mortality, reducing by 42% from 43 per 1,000 live births in 1988 to 25 per 1,000 live births in 2017, compared to under-five mortality which reduced by 67% from 155 per 1,000 live births in 1988 to 52 per 1,000 live births in 2017 (19). In fact, between 2007 and 2018, Ghana’s neonatal mortality rate reduced by only 14% from 29 to 25 deaths per 1,000 live births (19). (Lines 66-70)

Methods 

Context: Line 103-104, in your sentence you mentioned "...qualitative and quantitative health". Please rephrase this to make it easier to understand. The first paragraph showing the context provided a brief background or the motivation for this study. However, in the discussion, have you considered comparing your results to those reported by the MEBCI project? What data quality issues did they report compared to your findings? In this paragraph, consider also specifying the initial data quality concerns/ issues observed or reported before this study. Also, did the MEBCI publish any paper relating such data quality issues? If so, incorporate such findings as part of the context section.

Response: We thank the reviewer for the suggestions. The sentence ‘The evaluation of MEBCI comprised three components; cost analysis, qualitative and quantitative health.’ has been omitted. 

We observed during preliminary data analysis of cause of neonatal mortality that some listed causes of death (COD) diagnosis such as HIV-exposed baby, Hepatitis B infection, and home delivery could physiologically not have caused the neonatal mortality. The MEBCI project did not officially publish the earlier findings, but rather sought to quantify the extent of the data quality issues, necessitating this study. The methods section has been modified to provide readers more information about the context for the study. 

 ‘During preliminary analysis of the COD data, concerns about some listed neonatal COD diagnoses such as HIV infection and Hepatitis B infection arose. This study was undertaken due to concerns about the quality of the data following this preliminary analysis. (Lines 109-111)

Line 121: A multistage sampling was used to sample what?

Response: We thank the reviewer for drawing our attention to this. We have reviewed the sentence and reads; ‘A multi-stage sampling technique was used to sample IND. (Line127)

Data collection and analysis: Consider splitting data collection and data analysis sections. Also, to enable understanding the LQAS methodology, I recommend you have a section of "Sample size and Sampling" where the second paragraph of the context section will fit better. Within this new section you also provide the formula used for sample size calculation (if used) as well as the threshold value(s) as per the LQAS theory. This is very important to readers who never heard of this methodology. After providing the sample size information, then describe the assumptions of LQAS and how it was applied as described in the first paragraph of data collection and analysis.

Response: We are grateful to the reviewer for bringing our attention to these. We have created separate sections for ‘Sample size and sampling’, ‘Data collection’ and ‘Data analysis’. No formula was used for sample size estimation. 

We have also provided under the newly created section ‘Sample size and Sampling’ the assumption underlying the LQAS method and a summary of its application (Lines 114-123). 

A detailed sampling and data collection process for the LQAS has been described in ‘Sample size and Sampling’ and ‘Data collection’ sections (Lines 125-152). 

Under the section ‘Data analysis, we have described how the decision rule for LQAS was applied in our study (Lines 155-170). 

The information on the second paragraph of data collection and analysis section can go to the section describing "Data collection".

Response: We are grateful to the reviewer for the comment. The paragraph ‘Data abstraction was done by a public health physician with clinical ………… If no certificate was issued, the diagnosis was compared with that in either the clinician’s notes or neonatal death audit record.’ has been moved to the section describing ‘Data collection’. (Lines 142-152) 

Line 146-148: Data accuracy was set at 95% and further described as 17/20 correct match to mean accuracy. Someone would imply 17/20 as 85% instead of 95%. To avoid confusion, please rephrase this sentence and provide a more clear description of the value of accuracy used in relation of 17/20 correct matches.

Response: Under the newly created ‘Data analysis’ section, a clearer description of how the decision theory was applied has been provided and is provided below. 

‘The diagnosis in the A&D register was said to be accurately matched to that in the validation source if the two diagnoses were consistent. Within a lot of twenty sampled IND (n = 20) with Xi accurately matched diagnoses and decision rule ‘d’, if Xi ≥ d, then the lot was classified as having accurate COD data. If Xi ˂ d, then the lot would have failed to achieve the target of having accurate COD data. The threshold for accurate data was set at 95%. According to the Lemeshow and Taber’s LQAS table, for a target of 95% and sample size of 20, the decision rule ‘d’ is 17, meaning a lot would have a minimum of 17 accurately matched diagnoses to have accurate COD data. Decision rule: If Xi ≥ 17, the lot has accurate cause of neonatal death data; Xi ˂ d, the lot does not have accurate cause of neonatal death data.’ (Lines 155-163) 

Line 148-149: You stated that. "Data from the lots were aggregated to provide weighted estimates of accuracy for the regions and years." Did you actually weight the data? How was this done? Provide a description in the data analysis section.

Response: We thank the reviewer for the feedback. Under the newly created ‘Data analysis’ section, we have described how weighting was done to obtain the weighted accuracy estimates.

‘Data across the lots were aggregated to provide overall, regional-level and yearly weighted estimates of data accuracy. Weighting was necessary to adjust for the differences in total facility and yearly IND burden and was done using the direct adjustment method. Lots were weighted according to their proportionate contribution to the total IND burden. For each lot, the proportion of accurately matched diagnoses was multiplied by the weight (number of IND for the lot divided by the total number of IND) to obtain the weighted proportion of accurately matched diagnoses for the lot. The overall, regional-level and yearly weighted accuracy estimates were then calculated by summing up the individual weighted estimates of involved lots.’ (Lines 164-171) 

Results 

For all figure citations within the text, I think the journal recommends citing as Fig 1. Fig 2

Response: We thank the reviewer for the correction. We have appropriately renamed the figures in the text. (Lines 148 and185) 

The descriptions related to Table 1 are summary estimates. I recommend you combine Table 1 and 2, i.e., Table 1 to be summarised in Table 2. Otherwise, summarise the information about the number of lots in Table 2 for ease of reference with the number of lots with accurate data. It is important to ensure such descriptions are reflected in your table(s).

Response: The comment is well noted. We have combined the tables 1 and 2 and created a new column in the new table ‘Table 1’ for ‘number of eligible lots’ to facilitate reference. (Line 202) 

The columns in Table 2 are about validated with death certificates and clinician’s note or audit report. However, the title is about "Death certificate completion..." which I think does not reflect the content within this table. Please consider rephrasing to reflect the actual information. Likewise, lines 173-180 the descriptions are more on the death certificates issued while the results in the table are about the percentages (and numbers) of IND validated against the source documents. Please revise.

Response: The column headings in formerly Table 2, now Table 1 have been revised accordingly to reflect the information in the title and text. (Line 202) 

 The last column of table 2, how were these percentages calculated. Please describe that in the methods section (data analysis) and/ or below this table.

Response: We thank the reviewer for the comment. We have described under the newly created ‘Data analysis’ section, how we calculated the weighted accuracy estimates.

‘Data across the lots were aggregated to provide overall, regional-level and yearly weighted estimates of data accuracy. Weighting was necessary to adjust for the differences in total facility and yearly IND burden and was done using the direct adjustment method. Lots were weighted according to their proportionate contribution to the total IND burden. For each lot, the proportion of accurately matched diagnoses was multiplied by the weight (number of IND for the lot divided by the total number of IND) to obtain the weighted proportion of accurately matched diagnoses for the lot. The overall, regional-level and yearly weighted accuracy estimates were then calculated by summing up the individual weighted estimates of involved lots. (Lines 164-171) 

Table 3: Consider including the most common causes of IND that were correctly matched. The reason for this is because, it would be even more informative to understand the common causes of neonatal deaths in Ghanaian health facilities. Correct matches is also a part of data quality (i.e., high accurate data). So, I think these should also be reported.

Response: The comment is well noted. We have included in the table, now Table 2, common cause of death (COD) diagnoses that were correctly matched. (Line 216) 

Please cite the respective table number where the results which are below Table 3 are described.

Response: We thank the reviewer for drawing our attention to this. We have appropriately cited the respective table number. 

‘… Except for 2016, which had a little more than 50% of lots meeting the target, other years had less than 50% of lots meeting the target (Table 1). (Line 220-221)

Discussion

The sentence in Line 242-243 "Medical COD certificates also serve many purposes other than for death registration." Specify the purposes for Medical COD certificates. This would also justify your conclusions. Please consider citing other literature (if any) that may support such purposes

Response: We are grateful to the reviewer for the feedback. We have revised the sentence to include the uses of medical cause of death certificates. 

‘Medical COD certificates also serve many purposes other than for death registration. They are an important source of routine cause of death data and may be used to determine the leading causes of death for health interventions. They may also be required as proof of death for claim to life insurance, pension benefits and inheritance.’ (Lines 261-264)

Conclusion

Based on the paragraph in line 226, A&D registers are important original source of routine health surveillance data. In your study, COD certificates were one of the reference documents to verify the causes of death data and therefore used to measure/ determine accuracy. My question is, why conclude COD certificates as unreliable, while also their data are not entered in the national system? Please consider this and revise your conclusion also in the abstract.

Response: The reviewer’s comment is well noted. Both A&D registers and medical cause of death certificates are important sources of routine mortality data. A&D registers are usually completed during the initial stage of hospital admission when investigations may be pending, thus a provisional diagnosis may be entered. The register may then be updated with the final diagnosis upon discharge of patient from the hospital or death of the patient. The medical cause of death certificate, which contains information on the immediate and underlying cause of death is completed by the last attending medical practitioner upon death of the patient. The death certificate is considered a more accurate source of COD data of the two because the most current diagnosis is used, thus the decision by authors to use the death certificates as the gold standard. In our study, we found out only 22% of sampled neonatal deaths had death certificates filled, thereby our conclusion that they were not reliably completed in a sample of health facilities. Ghana presently does not have a functional electronic national database for health facilities to enter their cause of death diagnoses. 

References

Please verify and correct some of your references such as number 27, 37 and 38.

Response: We take note of this important comment. All references have been crosschecked and appropriately formatted to meet journal guidelines.

---

## [Decision Letter · Decision Letter 1]

31 Aug 2020

Evaluation of neonatal mortality data completeness and accuracy in Ghana

PONE-D-20-01438R1

Dear Dr. Dadzie,

We’re pleased to inform you that your manuscript has been judged scientifically suitable for publication and will be formally accepted for publication once it meets all outstanding technical requirements.

Kind regards,

Ricardo Q. Gurgel, PhD

Academic Editor

PLOS ONE

Additional Editor Comments (optional):

Reviewers' comments:

Reviewer's Responses to Questions

**Comments to the Author**

1. If the authors have adequately addressed your comments raised in a previous round of review and you feel that this manuscript is now acceptable for publication, you may indicate that here to bypass the “Comments to the Author” section, enter your conflict of interest statement in the “Confidential to Editor” section, and submit your "Accept" recommendation.

Reviewer #1: All comments have been addressed

Reviewer #2: All comments have been addressed

2. Is the manuscript technically sound, and do the data support the conclusions?

Reviewer #1: Yes

Reviewer #2: Yes

3. Has the statistical analysis been performed appropriately and rigorously? 

Reviewer #1: Yes

Reviewer #2: Yes

4. Have the authors made all data underlying the findings in their manuscript fully available?

Reviewer #1: Yes

Reviewer #2: Yes

5. Is the manuscript presented in an intelligible fashion and written in standard English?

Reviewer #1: Yes

Reviewer #2: Yes

6. Review Comments to the Author

Reviewer #1: Response 1 : Evaluation of neonatal mortality data completeness and accuracy in Ghana

Comment : Can omit “ In Ghana” which can be included in methodology section

Response 2 ( Line 23 ): Seems OK

Response 3 ( Line 104) : seems Ok

Response 4 : Ok

Response 5 : done as per comments

Reviewer #2: All the initial comments have been addressed. I have some minor issues for your consideration;

Abstract, please write the numbers as n (%) instead of (n, %).

Methods, data analysis section line 156, describe what the notation Xi represent. If it represent lots, be specific for clarity.

Results, Table 1 on line 203, indicate whether the % IND and n (%) in the last two columns were weighted. Either start with the word "weighted" or put a not below the table.

Also, why did you chose to delete the column on %IND validated with the clinician's note or audit report? Were these not an important findings? Because clinician's note or audit report was one of the validation sources, consider incorporating such results in the results and discussion sections.

7. PLOS authors have the option to publish the peer review history of their article (what does this mean?). If published, this will include your full peer review and any attached files.

Reviewer #1: No

Reviewer #2: No

---

## [Editor Report · Acceptance letter]

3 Sep 2020

PONE-D-20-01438R1 

Evaluation of neonatal mortality data completeness and accuracy in Ghana 

Dear Dr. Dadzie:

I'm pleased to inform you that your manuscript has been deemed suitable for publication in PLOS ONE. Congratulations! Your manuscript is now with our production department. 

Kind regards, 

on behalf of

Professor Ricardo Q. Gurgel 

Academic Editor

PLOS ONE